# Assessment of Interleukin-15 (IL-15) Concentration in Children with Idiopathic Nephrotic Syndrome

**DOI:** 10.3390/ijms24086993

**Published:** 2023-04-10

**Authors:** Andrzej Badeński, Marta Badeńska, Elżbieta Świętochowska, Artur Janek, Aleksandra Gliwińska, Aurelia Morawiec-Knysak, Maria Szczepańska

**Affiliations:** 1Department of Pediatrics, Faculty of Medical Sciences in Zabrze, Medical University of Silesia in Katowice, ul. 3 Maja 13/15, 41-800 Zabrze, Poland; 2Department of Medical and Molecular Biology, Faculty of Medical Sciences in Zabrze, Medical University of Silesia in Katowice, ul. Jordana 19, 41-808 Zabrze-Rokitnica, Poland; 3Department of Pediatric Nephrology with Dialysis Division for Children, Public Clinical Hospital No. 1 in Zabrze, ul. 3 Maja 13/15, 41-800 Zabrze, Poland

**Keywords:** idiopathic nephrotic syndrome, IL-15, children, marker

## Abstract

Idiopathic nephrotic syndrome (INS) is a chronic glomerular disease in children, characterized by severe proteinuria, hypoalbuminemia, and/or presence of edema and hyperlipidemia. The pathogenesis, however, has not been yet established. The clinical course of the disease is characterized by frequent relapses. Interleukin-15 (IL-15) is a pro-inflammatory cytokine, that apart from its involvement in the immune system, was found to be playing a vital role in various cells’ functioning, including renal tissue. It is desirable to look for new predictors of INS. Our study aimed to evaluate IL-15 as a potential marker in the early diagnosis of the disease. The cohort participating in the study consisted of patients hospitalized in Clinical Hospital No. 1 in Zabrze, from December 2019 to December 2021, including study group with INS (*n* = 30) and control group (*n* = 44). Results: The concentration of IL-15 in both serum and urine was significantly elevated in patients with INS, compared to healthy controls. The cytokine might serve as a marker of the disease, however, further research on larger study groups is needed.

## 1. Introduction

### 1.1. The Idiopathic Nephrotic Syndrome

Idiopathic nephrotic syndrome (INS) is the most common chronic glomerular disease in children with an incidence of 15–16.9 per 100,000 children. It is characterized by severe proteinuria, hypoalbuminemia, and/or presence of edema and hyperlipidemia [1]. The clinical course of the disease is most often associated with relapses preceded by infectious events. The cause of the INS has not yet been established, however, the pathogenesis of this disease is said to be involving immune disorders, systemic circulatory factors, or hereditary structural abnormalities of the podocyte [1]. Anders et al. recently suggested a new classification of glomerulonephritis, based on immunopathophysiology, so as to integrate the clinical experience and the growing number of available immunotherapies better. Such an approach may also be helpful in understanding the mechanisms of INS [2].

Diagnosis is established based on clinical symptoms thus kidney biopsy is not performed unless steroid resistance is observed [3]. Most young patients respond well to steroid treatment. As a result of daily oral treatment with prednisone, complete remission of proteinuria is observed in approximately 85% of INS cases [4]. Such a type of INS is referred to as steroid-sensitive nephrotic syndrome (SSNS). The lack of remission after 4–6 weeks of treatment classifies the disease as steroid-resistant NS (SRNS). Genetic disorders related to the disease are more frequently described in children diagnosed with SRNS [5].

### 1.2. Interleukin-15

Interleukin-15 (IL-15) is a pro-inflammatory cytokine with the ability to induce proliferation of T cells, to enhance T and natural killer cell cytotoxicity and to protect T cells and neutrophils from apoptosis. It is also able to stimulate secretion of other pro-inflammatory cytokines [6]. The IL-15 gene has been detected in the placenta, skeletal muscle, kidney, lung, and heart [7]. Apart from the impact on the immune system, IL-15 is said to be a survival factor for various cell types, such as hematopoietic cells, but also keratinocytes, fibroblasts, hepatocytes, and epithelial cells [6,8,9,10].

Up to this point, several forms of IL-15 have been mentioned in the literature, including two soluble types: the monomeric IL-15 secreted by accessory cells and IL-15/IL-15Rα complex [11,12,13,14] as well as two membrane-connected forms: the membrane-bound form (mb-IL-15), anchored at the cell membrane using the IL-15Rα chain, and the transmembrane form (tmb-IL-15), independent from IL-15R [12,15,16,17] (Figure 1). Studies have shown that physiological activity of the cytokine is mainly performed by IL15/IL-15Rα complex and tmb-IL-15 [11,18,19,20]. When analyzing forms detected in a blood serum, concentration of the cytokine by itself was minimal, whereas the majority was represented by IL-15/IL-15Rα. Moreover, co-expression of IL-15 and its receptor is said to be stabilizing both molecules, resulting in accumulation of the complex on the cell membrane or its excretion. Circulating IL-15/IL-15Rα complexes are described as highly effective lymphocyte activation factors. Their effect is mostly acknowledged on CD8-positive T cells, and slightly less on CD4-positive T cells and natural killer cells (NK) [11,21]. The above-mentioned coexistence and collaborative functioning of IL-15 and its receptor suggests that IL-15Rα should be considered as a significant contribution to an active cytokine [11].

It is desirable to seek new methods for the early detection of chronic diseases, including INS. Our study aimed to evaluate IL-15 as a potential marker in the early diagnosis of INS disease and as a predictor of proteinuria.

## 2. Results

### 2.1. Descriptive Analysis and Comparison—INS Patients and Control Group

Contrary to the control group, patients suffering from INS had proteinuria with mean urine protein level at 4.45 ± 8.78 g/L. However, an increase in serum creatinine levels was detected neither in the control group, nor the study group, therefore, the mean creatinine level for all patients reached 34.00 ± 14.60 umol/L. The majority of patients with INS (23 individuals, 76.7%) were participating in the study during a relapse, whereas for the remaining 7 patients (23.3%), the diagnosis had been just established. The treatment mainly included steroid therapy (19 cases), however, one case required cyclophosphamide and 10 patients (33.3%) were receiving cyclosporine. Features of the groups, both control and study, are presented in Table 1. Table 2 presents selected laboratory parameters for the study group.

### 2.2. IL-15—Results

Student’s t-test for independent samples was used to establish the possible reliance between IL-15 concentration in serum and urine and the manifestation of INS. In the study group, a significantly higher concentration of IL-15 in urine (*p* < 2.2 × 10^−16^) and in serum (*p* = 9.6 × 10^−10^) were noted, compared to the control group (Figure 2).

Considering the link between serum and urine concentrations of IL-15, the R-Pearson’s correlation coefficient was applied separately for both groups. No significant positive or negative correlation were confirmed in the groups (Figure 3).

Moreover, the suspicion of the influence of the ongoing therapies was raised, therefore, patients on cyclosporine were assessed separately. Nevertheless, significant differences in serum or urine IL-15 concentrations among INS patients on cyclosporine compared to those on other medications were not found (Figure 4).

Furthermore, a serum IL-15 to serum creatinine ratio was established for both groups. The mean parameter for the study group was 0.3, whereas for the control group was much lower, reaching 0.08 (Table 3). Other significant correlations, considering age, body weight, blood pressure, total serum protein, cholesterol levels, c-reactive protein, and other laboratory parameters, were not found.

## 3. Discussion

The pathogenesis of INS in children remains unclear. Due to the immunological involvement in the course of the disease, the role of pro-inflammatory cytokines is suspected. Therefore, this study’s design was based on a hypothesis, that IL-15 as a promotor of inflammation could act as a potential marker for INS. Indeed, the outcome of our research shows a significant increase in serum and urine IL-15 concentrations among patients with INS, compared to healthy controls. However, IL-15 level correlated neither with crp value, serum total proteins nor proteinuria.

IL-15 is a cytokine reported in the literature quite often. As for its impact on kidney function, up to this point, there has been a variety of data supporting different approaches in this area. Among various cell types proven to produce diverse forms of IL-15, renal tubular epithelial cells (TEC) are said to excrete IL-15 as a significant factor of immune stability and biological functioning of the renal epithelium [23]. The presence of all three receptor subunits, α, β, and γ, has been described in TEC in several studies [24,25]. A signaling cascade, involving the IL-15Rγ/JAK3 pathway, has been reported to maintain epithelial phenotype, and was found in normal human renal proximal tubular epithelial cells (RPTEC) [26]. According to literature, monomeric IL-15, secreted by kidneys in low concentrations, is functioning through autocrine-paracrine and/or juxtracrine loops as a cell survival agent [24,27,28]. On the other hand, mb-IL-15 form was also discovered in kidney cells, and some authors considered it to play a major role in renal homeostasis mechanism [23]. In our study, higher concentrations of IL-15 in children with INS may indicate its role in maintaining the kidney function. Moreover, IL-15/IL-15Rα complex may also be produced and excreted by the renal epithelial cells, resulting in promoting local activity of IL-15 with the participation of the neighboring tissues or leukocytic infiltrations in the area [29].

Prolonged survival of the renal epithelial cells is said to be triggered by anti-apoptotic mechanisms of IL-15 signaling. It is reported that basic levels of this cytokine and its receptor in the kidneys raise the apoptotic threshold, particularly in reference to the apoptosis induced by T cells. Presumably, another protective mechanism is based on elimination of macrophages and CD4-positive T cells in certain immunological pathologies. Moreover, it was discovered that IL-15 deficiency led to exaggerated TEC apoptosis. Therefore, in kidney pathology, IL-15 may serve as renal cells’ damage protector [24,29]. Devocelle et al. presented that in kidney biopsies from patients with different nephropathies (including IgA nephropathy, segmental, and membranous glomerulonephritis), lower IL-15 expression, compared to healthy controls, was documented, and the same results were shown on protein level [30]. Mooslechner et al. highlighted that homeostatic levels of IL-15 maintain kidney health; therefore, it could be considered as a potential candidate for immunotherapy. In their research, a mouse model of nephrotoxic serum nephritis (NTS) was used, which was intended to refer to glomerulonephritis in humans. A low-dose IL-15 immunotherapy resulted in an improvement in kidney function with reduced albuminuria and prolonged cell survival in mice. While high levels of IL-15 were considered to stimulate the immune system, a significant reduction of a cytotoxic activity of CD8-positive T cells was detected as a response to low IL-15 doses [31].

In our study, the increased level of IL-15 was not reduced during cyclosporine therapy—it remained on a high level, contrary to the above-mentioned experimental studies on animals.

Still, IL-15 belongs to a group of pro-inflammatory cytokines, therefore, its possible pathological impact on renal cells was explored as well. The connection between acute kidney transplant rejection and increased expression of IL-15 and IL-15Rα was discovered and confirmed in several studies, providing biopsies of the rejected kidney transplants [32] and suggesting a positive role of IL-15 inhibitors in prolonged renal transplant survival [18]. Hingorani et al. found that levels of IL-6, IL-15, and CC-chemokine ligand 2 in urine were associated with the risk of developing proteinuria as a marker of kidney injury after bone marrow transplantation [33]. Moreover, some therapeutic strategies for T cell-mediated autoimmune inflammatory diseases consider the use of IL-15 inhibitors [31]. Additionally, an impact of IL-15 on a progression of renal cancer is suspected, as particular isoforms of this cytokine act as a down-regulator of E-cadherin, resulting in increased risk of cell neoplastic transformation and tumor growth [17,34]. A recent study on lupus nephritis provided data supporting a theory that IL-15 could be a major factor for enhancement of cytotoxic activity of CD4-positive CD28-negative T cells. It would result in their activation, adhesion, and increased cytokine secretion in the renal tissues. Elevated IFN-γ concentration and apoptosis were observed as an IL-15 impact on the above-mentioned lymphocytes [35]. In Pacheco-Lugo et al.’s study, lupus nephritis patients had increased plasma IL-15 levels as compared to patients with no renal involvement [36]. These authors pointed out that there was a significantly high level of IL-15 levels among patients with kidney symptoms, which would be in compliance with our results and in comparison, of children with INS to healthy controls. Additionally, Liang Li et al. highlighted the fact that IL-15 significantly promoted tissue-resident-memory T cell formation in kidneys and their activation, thereby enhancing podocyte injury and glomerulosclerosis [37].

### Limitations of the Study

The study group size seemed to be a limitation for our study. Our analysis included ROC curves to assess sensitivity and specificity of the IL-15 as a potential marker. However, the area under the curve (AUC) reached 100%, which would indicate that there is a need for larger study groups to obtain more reliable outcome. Further assessments on larger groups of patients should be conducted.

## 4. Materials and Methods

### 4.1. Studied Groups

Patients representing the study group were 2 to 17 years old (*n* = 30). They stayed under the care of the Department of Pediatric Nephrology with the Subdivision of Dialysis at the Clinical Hospital No. 1 in Zabrze, Medical University of Silesia in Katowice, from December 2019 to December 2021, due to idiopathic nephrotic syndrome.

Inclusion criteria for this group comprised of confirmed diagnosis of INS, patient newly diagnosed or in relapse of underlying disease more than 3 months, and less than 18 years of age. The exclusion criteria were the following: non-nephrotic proteinuria, congenital NS and NS secondary to metabolic, infectious, vascular, malignant, and cardiac diseases.

The diagnosis of idiopathic nephrotic syndrome was established based on the Recommendations of Polish Society for Pediatric Nephrology for the management of children with nephrotic syndrome from 2015 [38]. We assembled medical history of all patients, including reports of INS relapses, ongoing therapies, and coexisting diseases.

Patients representing the control group (*n* = 44) were admitted at the same period of time to the Department of Pediatric Nephrology with the Subdivision of Dialysis due to bedwetting or to the Department of Surgery of Child Developmental Defects and Traumatology of the Clinical Hospital No.1 in Zabrze, Medical University of Silesia in Katowice for procedures as part of one-day surgery. No patient from this group was described to have impaired kidney function, nor chronic or infectious disease.

All 74 patients were included into the research, regarding inclusion criteria, and then divided into study (*n* = 30) and control (*n* = 44) group; 14 girls (46.6%) and 16 boys (53.4%) with a mean age of 7.67 ± 4.14 years were representing the study group. The control group with 18 girls (40.9%) and 26 boys (59.1%) had their mean age established at 7.75 ± 4.10 years.

This research project was approved by the Ethics Committee of the Medical University of Silesia in Katowice (PCN/0022/KB1/133/19). A written informed consent was obtained from caregivers of all the children and, in the case of participants older than 16 years, also from the child.

### 4.2. Laboratory Tests

Complete blood count and biochemical tests (serum urea, serum creatinine, serum uric acid, cholesterol level, level of triglycerides, total serum protein concentration, blood electrolytes, blood gas test, urinalysis) were performed in both groups of patients. Classic Schwartz formula with the age-appropriate k-coefficient amendment was used to calculate the estimated glomerular filtration rate (eGFR) (mL/min/1.73 m^2^) [39].

### 4.3. IL-15 Concentration

The concentration of IL-15 in blood serum and urine was performed by ELISA technique with the use of R&D Systems (USA) kit-Human IL Immunoassay Cat. No. D1500. The samples were tested 3 times and the final result was a mean value. The analytical procedure was in accordance with the technological instructions attached by the manufacturer to the kit. Absorbance readings were taken on a SYNERGY/H1 reader (BioTek, Santa Clara, CA, USA) at a wavelength of 450 nm using a reference wavelength of 570 nm. The elaboration of the results was performed using the Gen5 v 3.05 computer program (BioTek, Santa Clara, CA, USA). The sensitivity of the method was 0.2 pg/mL. The precision of the method in the simultaneous series (imprecision) was 5.1%.

### 4.4. Anthropometric Measurements

Anthropometric parameters, such as weight, presented in kilograms, and height, given in centimeters, were noted for both groups (expressed to two decimal places). Each individual had their Body Mass Index (BMI) calculated using the equation: weight/height^2^ (kg/m^2^) and blood pressure was taken. Percentile charts from OLA and OLAF studies for Polish children [40,41] were used to plot the above-mentioned parameters for age and sex. Moreover, SDS values were calculated, providing comparison of those measurements between the groups.

No data in the collected groups were missing.

### 4.5. Statistical Analysis

Statistical analysis was conducted with the use of the R Studio software. Means with standard deviation or medians with quartile range were used in descriptive statistics. The Shapiro–Wilk test was applied to establish normal distribution of the assessed parameters. Homogeneity of variance was evaluated by the Levene’s test. The Pearson’s correlation coefficient indicated correlations between the parameters. The t-test for independent variables was used to compare selected criteria in both groups, study and control. Statistical significance was set at *p* value < 0.05.

This research was conducted as a part of a series of studies, performed using the same population [22], therefore, description of the groups is analogical to the studies from the series.

## 5. Conclusions

Our study provides the information that the concentration of IL-15 in both serum and urine was significantly elevated in pediatric patients with INS in the active phase of the disease, therefore, it could serve as a marker of the severity of the disease. However, the literature shows different approaches in the expected influence of IL-15 on the kidney function. Therefore, the cause of this cytokine and increased secretion in INS is not established, which was confirmed by us by the lack of correlations with other parameters. To deliver more data in this field, further research using larger study groups is required.

## Figures and Tables

**Figure 1 ijms-24-06993-f001:**
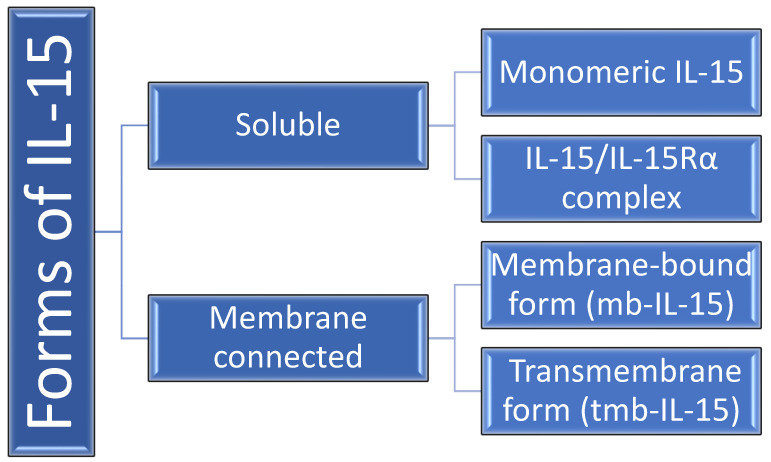
Forms of IL-15 in humans. IL-15Rα–IL-15 receptor α.

**Figure 2 ijms-24-06993-f002:**
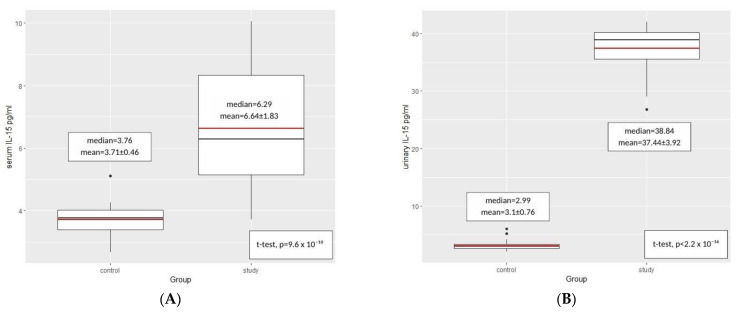
IL-15 concentration comparison between the control group and the study group in serum (**A**) and urine (**B**) (student’s t-test for independent samples). Red line represents mean value and black line represents the median.

**Figure 3 ijms-24-06993-f003:**
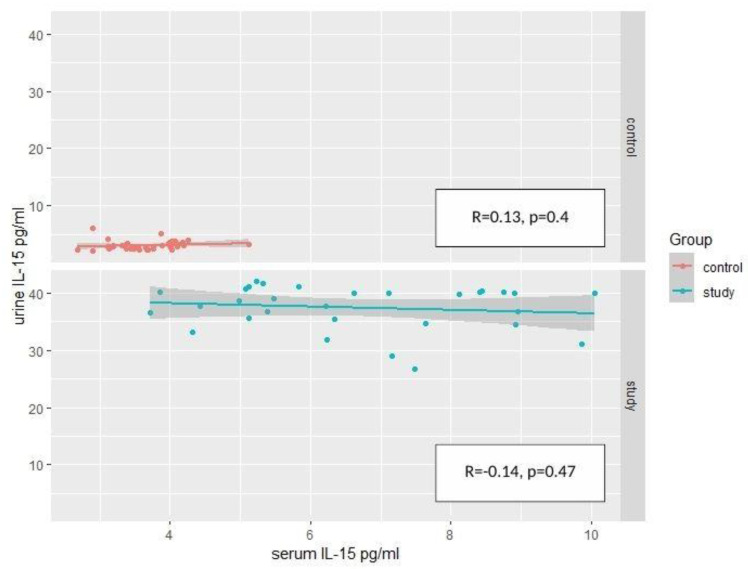
Correlations between the concentration of IL-15 in the serum and urine, considering the division into the study group and the control group (R-Pearson’s correlation coefficient). The test probability of both correlations is higher than the adopted significance level α = 0.05).

**Figure 4 ijms-24-06993-f004:**
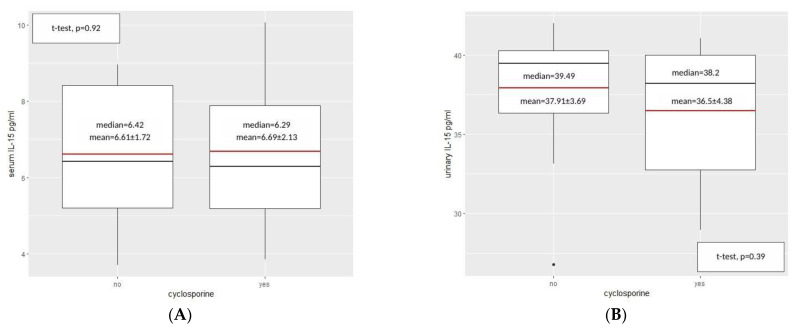
IL-15 concentration comparison in the study group in serum (**A**) and urine (**B**) between the patients on cyclosporine (yes) and treated with different drug (no), (Student’s t-test for independent samples). Red line represents mean value and black line represents the median.

**Table 1 ijms-24-06993-t001:** Characteristics of the groups (population described in the previous research) and their selected anthropometric and laboratory parameters [22].

Parameter	Children with Idiopathic Nephrotic Syndrome (*n* = 30)			Control Group (*n* = 44)		
	Whole Group	Female	Male	Whole Group	Female	Male
Age (year)	7.50 ± 4.14 (2.00–17.00)	6.14 ± 3.35 (2.00–13.00)	9.00 ± 4.40 (2.00–17.00)	7.75 ± 4.10 (2.00–17.50)	8.33 ± 3.95 (2.00–17.00)	9.85 ± 4.17 (4.50–17.50)
Height (cm)	125 ± 25.00 (92.00–179.00)	115.46 ± 19.24 (92.00–158.00)	134.41 ± 26.70 (92.00–179.00)	127 ± 24.90 (82.00–197.00)	127.50 ± 21.59 (82.00–170.00)	142.17 ± 25.60 (109.00–197.00)
SDS for height	0.06 ± 1.36 (−2.80–2.29)	−00.7 ± 1.20 (−2.80–2.20)	0.08 ± 1.53 (−2.80–2.29)	0.09 ± 1.10 (1.53–3.00)	−0.14 ± 1.08 (−1.53–2.35)	0.35 ± 1.08 (−1.34–3.00)
BW (kg)	29.80 ± 21.60 (12.10–88.90)	26.54 ± 13.55 (12.10–59.80)	43.26 ± 24.49 (14.50–88.90)	27.10 ± 18.80 (9.70–87.50)	30.35 ± 15.03 (9.70–71.00)	38.38 ± 20.68 (15.00–87.50)
SDS for BW	1.15 ± 2.1 (−1.83–9.58)	1.12 ± 1.58 (−1.83–3.80)	1.75 ± 2.49 (−1.63–9.58)	0.07 ± 1.11 (−2.34–2.12)	0.08 ± 1.23 (−1.87–2.11)	0.04 ± 1.04 (−2.34–2.12)
BMI (kg/m^2^)	18.90 ± 4.07 (13.50–30.90)	18.41 ± 3.20 (13.82–23.98)	20.92 ± 4.48 (13.50–30.90)	16.60 ± 3.45 (12.40–26.50)	17.52 ± 3.12 (13.90–24.60)	17.52 ± 3.72 (12.40–26.50)
SDS for BMI	0.83 ± 2.10 (−2.18–8.17)	0.07 ± 1.18 (−1.47–2.35)	2.20 ± 2.26 (−2.18–8.17)	−0.06 ± 1.18 (−2.97–2.29)	0.25 ± 1.05 (−1.38–2.29)	−0.25 ± 1.25 (−2.97–2.02)
SYS (mmHg)	115 ± 14.6 (81.00–152.00)	108.21 ± 14.87 (81.00–138.00)	117.88 ± 13.17 (102.00–152.00)	110.00 ± 11.30 (85.00–134.00)	107.17 ± 8.54 (89.00–122.00)	113.96 ± 12.25 (85.00–134.00)
DIA (mmHg)	67.50 ± 10.90 (50.00–90.00)	64.71 ± 9.51 (50.00–88.00)	73.44 ± 10.66 (56.00–90.00)	70.00 ± 12.00 (40.00–107.00)	66.83 ± 11.45 (45.00–96.00)	69.96 ± 12.44 (40.00–107.00)
MAP (mmHg)	82.50 ± 11.30 (67.00–111.00)	79.21 ± 9.97 (67.00–100)	88.25 ± 11.03 (71.33–111.00)	82.50 ± 10.70 (58.70–116.00)	80.28 ± 9.54 (62.33–102.33)	84.63 ± 11.23 (58.67–115.70)

Data are presented as mean ± standard deviation (minimum–maximum). BW: body weight; SDS: standard deviation score; BMI: Body Mass Index; SYS: systolic arterial pressure; DIA: diastolic arterial pressure; MAP: mean arterial pressure.

**Table 2 ijms-24-06993-t002:** Characteristics of the study group and their selected laboratory parameters.

Parameter	Children with Idiopathic Nephrotic Syndrome (*n* = 30)		
	Whole Group	Female	Male
C-reactive protein (mg/L)	2.88 ± 7.13 (0.12–31.69)	2.92 ± 6.60 (0.12–24.60)	2.84 ± 7.78 (0.2–31.69)
Total serum protein (g/L)	49.84 ± 9.15 (33.60–67.00)	50.29 ± 8.02 (39.50–66.80)	49.45 ± 10.28 (33.60–67.00)
Serum albumin (mg/mL)	26.42 ± 9.21 (11.24–41.4)	27.47 ± 9.39 (11.95–41.40)	25.51 ± 9.26 (11.24–40.31)
Urea concentration (mmol/L)	4.40 ± 1.90 (2.30–10.00)	3.65 ± 1.27 (2.40–5.90)	5.06 ± 2.14 (2.30–10.00)
Uric acid level (mg/dL)	284.37 ± 83.49 (139.00–453.00)	233.14 ± 61.33 (139.00–337.00)	329.19 ± 75.01 (163.00–453.00)
Serum creatinine (μmol/L)	34.10 ± 14.64 (14.00 ± 75.00)	27.36 ± 9.59 (15.00–47.00)	40.00 ± 15.97 (14.00–75.00)
Total cholesterol (mmol/L)	7.97 ± 2.97 (2.81–14.33)	7.44 ± 3.12 (2.81–14.33)	8.44 ± 2.84 (5.11–14.15)
Triglyceride level (mmol/L)	2.63 ± 1.88 (0.74–8.46)	2.40 ± 2.28 (0.74–8.46)	2.81 ± 1.54 (1.56–7.73)
Urine protein (g/L)	4.45 ± 8.78 (0.67–35.60)	6.04 ± 6.15 (0.88–23.70)	10.28 ± 10.35 (0.67–35.60)

Data are presented as mean ± standard deviation (minimum–maximum).

**Table 3 ijms-24-06993-t003:** Serum IL-15 to serum creatinine ratio for study and control group.

Parameter	Control group (*n* = 44)	Study group (*n* = 30)
Serum IL-15/serum creatinine	0.08 ± 0.02 (0.04–0.13)	0.30 ± 0.33 (0.05–1.88) *

Data are presented as the mean ± standard deviation (minimum–maximum). * *p* < 0.05.

## Data Availability

The data presented in this study are available on request from the corresponding author. The data are not publicly available due to privacy issues.

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
