# Peer review of "Assessment of Interleukin-15 (IL-15) Concentration in Children with Idiopathic Nephrotic Syndrome"

_ijms, 2023, doi:10.3390/ijms24086993_

Round 1

Reviewer 1 Report

The authors of the paper: "Assessment of interleukin-15 (IL-15) concentration in children with idiopathic nephrotic syndrome", present studies of children diagnosed with idiopathic nephrotic syndrome. In their work, the authors describe the results related to the level of IL-15 in these patients and suggest that the level of this interleukin could play a potential role of a biomarker molecule.

The manuscript is carefully written and contains no editorial errors. The presented research is interesting. However, after analyzing the work and the research done, I have a few questions for the authors and suggestions that I present below:

1. Table 2, please correct the units in the table from mg/l to mg/L as in the case of mg/dL

2. Urine protein level in the control group, was the amount of urine protein equal to 0 in all cases in the control group? Perhaps the authors will adopt a specific value at which they consider the result to be below the detection range?

3. The authors suggest the potential use of IL-15 as a marker molecule, are you able to prepare ROC curves?

4. How did the researchers determine the exclusion criterion for infections in the studied patients?

5. The IL-15 test was performed by ELISA, how many replicates? Perhaps the samples were only tested once, and if so, how do you ensure the reliability of the results?

6. Are the authors able to test IL-15 by other methods? Flow cytometry, RT-PCR? to confirm the results obtained by the ELISA method.

7. Are the authors able to find newer articles from the last 10 years on the presented subject matter?

Summing up, the work is the most valuable, and it can be seen that it has been carefully written, but in my opinion some issues should be clarified. Nevertheless, I congratulate the authors of the work.

Author Response

The authors of the paper: "Assessment of interleukin-15 (IL-15) concentration in children with idiopathic nephrotic syndrome", present studies of children diagnosed with idiopathic nephrotic syndrome. In their work, the authors describe the results related to the level of IL-15 in these patients and suggest that the level of this interleukin could play a potential role of a biomarker molecule.

The manuscript is carefully written and contains no editorial errors. The presented research is interesting. However, after analyzing the work and the research done, I have a few questions for the authors and suggestions that I present below:

  1. Table 2, please correct the units in the table from mg/l to mg/L as in the case of mg/dL
    1. Answer: The corrections have been made.

  2. Urine protein level in the control group, was the amount of urine protein equal to 0 in all cases in the control group? Perhaps the authors will adopt a specific value at which they consider the result to be below the detection range?
    1. Answer: Thank you for your suggestion. Urine protein level in control group for all patients showed 0. The corrections in the tables were made.

  3. The authors suggest the potential use of IL-15 as a marker molecule, are you able to prepare ROC curves?
    1. Answer: Thank you for your suggestion. Our statistical analysis included ROC curves; however, the outcome of AUC was 100%. Therefore, we decided not to present the outcome in our manuscript, as there is a need to perform the analysis on larger study groups.
    2. How did the researchers determine the exclusion criterion for infections in the studied patients?
      a. Answer: Thank you for your question. Patients suffering from nephrotic syndrome originating from infectious disease weren’t taken into consideration for our study, as we examined children with idiopathic nephrotic syndrome (INS). On admission, the patients were carefully examined and inflammatory parameters were assessed in blood samples.
    3. The IL-15 test was performed by ELISA, how many replicates? Perhaps the samples were only tested once, and if so, how do you ensure the reliability of the results?
      a. Answer: The samples were tested 3 times and the final result was a mean value.
    4. Are the authors able to test IL-15 by other methods? Flow cytometry, RT-PCR? to confirm the results obtained by the ELISA method.
      a. Answer: Thank you for your question. Unfortunately, in our grant, no other tests were available.
    5. Are the authors able to find newer articles from the last 10 years on the presented subject matter?
      a. Answer: Thank you for the question. We did our best to collect the most recent data on the topic. We have noticed, with great surprise, that there are not many current articles and we included the available studies in our manuscript.

Summing up, the work is the most valuable, and it can be seen that it has been carefully written, but in my opinion some issues should be clarified. Nevertheless, I congratulate the authors of the work.

Reviewer 2 Report

Authors present an observational, analytical, cohort, single centre study to evaluate a possible link between idiopathic nephrotic syndrome (INS) and serum-urine levels of interleukin-15 (IL-15). They found higher concentration of IL-15 in serum and urine of the study group and supposed IL-15 could be considered a marker of diagnosis and prognosis of  INS.

This is an interesting article but some revisions are needed before the pubblication.

The iconographic support must necessarily be implemented.

Firstly we suggest to change the order of sections for a better understanding of the article (introduction, materials and methods, results, discussion, conclusions).

In the introduction section, some changes are recommended:

-At lines 46-47, diagnosis of INS should precede therapy

-At line 54 authors should add "it is also annunced to stimulate secretion of OTHER proinflammatory cytokines"

- From lines 59 to 76 authors describes the different forms of IL-15. They should include a summary figure in this section in order to semplificate the text.

In the results section, some changes are recommended:

- We suggest to move lines from 80 to 83 to "material and methods section" to better qualify the two group's population.

- Table 1 contains anthropometic parameters while Table 2 contains laboratory parametre. "urine protein (g/l) should be in Table 2.

In the materials and methods section, some changes are recommended:

- Authors shoud better explain that the study group's and control group's populations were screened amoung patients ammetted in the Clinical Hospitaly from December 2019 to Decembre 2021.

- Inclusion and exclusion criteria should be better explain. Active or quiescent disease? What laboratory parameters were considered to define a "impaired kidney function?

Author Response

Authors present an observational, analytical, cohort, single centre study to evaluate a possible link between idiopathic nephrotic syndrome (INS) and serum-urine levels of interleukin-15 (IL-15). They found higher concentration of IL-15 in serum and urine of the study group and supposed IL-15 could be considered a marker of diagnosis and prognosis of INS.

This is an interesting article but some revisions are needed before the publication.

The iconographic support must necessarily be implemented.

Firstly, we suggest to change the order of sections for a better understanding of the article (introduction, materials and methods, results, discussion, conclusions).

Answer: Thank you for your suggestion. However, the journal prefers the presented structure of the manuscript.

In the introduction section, some changes are recommended:

-At lines 46-47, diagnosis of INS should precede therapy.
            Answer: The corrections have been made.

-At line 54 authors should add "it is also announced to stimulate secretion of OTHER proinflammatory cytokines"
            Answer: The line was corrected.

- From lines 59 to 76 authors describes the different forms of IL-15. They should include a summary figure in this section in order to simplify the text.

            Answer: Thank you for your suggestion. A suitable figure was added to the text.

In the results section, some changes are recommended:

- We suggest to move lines from 80 to 83 to "material and methods section" to better qualify the two group's population.

            Answer: Thank you for the suggestion, the corrections were made.

- Table 1 contains anthropometic parameters while Table 2 contains laboratory parametre. "urine protein (g/l) should be in Table 2.
            Answer: Thank you for that note. The tables were corrected.

In the materials and methods section, some changes are recommended:

- Authors should better explain that the study group's and control group's populations were screened among patients ammetted in the Clinical Hospital from December 2019 to Decembre 2021.

            Answer: Thank you for your suggestion. Patients with INS admitted to our hospital from December 2019 to December 2021 were always taken into consideration to be a part of our study as a study group. The control group were admitted at the same period of time to the Department of Pediatric Nephrology with the Subdivision of Dialysis due to bedwetting or to the Department of Surgery of Child Developmental Defects and Traumatology.

- Inclusion and exclusion criteria should be better explain. Active or quiescent disease? What laboratory parameters were considered to define a "impaired kidney function?

Answer: The inclusion criteria include patients newly diagnosed or in a relapse, which indicates patients in active disease. The diagnosis of idiopathic nephrotic syndrome was established based on the Recommendations of Polish Society for Pediatric Nephrology for the management of children with nephrotic syndrome from 2015. Kidney function was assessed using Classic Schwartz formula with the age-appropriate k-coefficient amendment to calculate the estimated glomerular filtration rate (eGFR) [ml/min/1.73m2].

Round 2

Reviewer 1 Report

Thanks to the authors for preparing the answers,

they are understandable, but I would like to ask you to include information from the answers in the main text of the manuscript. In particular, for information on the number of repetitions for the ELISA test.

However, as for the ROC Curves, this information could also be included in the description of the limitations for the performed tests.

After making these changes, I see no obstacles in publishing the results obtained.

Author Response

Thanks to the authors for preparing the answers,

they are understandable, but I would like to ask you to include information from the answers in the main text of the manuscript. In particular, for information on the number of repetitions for the ELISA test.

Answer: Thank you for the suggestion, we added the information.

However, as for the ROC Curves, this information could also be included in the description of the limitations for the performed tests.

Answer: Thank you for the suggestion, we added the information.

After making these changes, I see no obstacles in publishing the results obtained.

Reviewer 2 Report

Authors have modified the article according to our suggestions. Now it is ready for publication. 

Author Response

Thank you very much.